A novel role for apatinib in enhancing radiosensitivity in non-small cell lung cancer cells by suppressing the AKT and ERK pathways

Li Lin 1 2
Li Yuexian 3
Zou Huawei 2 Zouhw999@163.com
1 The First Oncology Department, The Fourth Hospital of China Medical University , Shenyang, Liaoning , China
2 Department of Oncology, Shengjing Hospital of China Medical University , Shenyang, Liaoning , China
3 Department of Radiation Oncology, Cancer Hospital of China Medical University, Liaoning Cancer Hospital & Institute , Shenyang, Liaoning , China
Uversky Vladimir
Electronic publication date: 2021 Oct 28
Publication date: 2021
Volume: 9
Electronic Location ID: e12356
Received 2021 May 18; Accepted 2021 Sep 30
Copyright: © 2021 Li et al.
Copyright year: 2021
Copyright holder: Li et al.
License: This is an open access article distributed under the terms of the Creative Commons Attribution License, which permits unrestricted use, distribution, reproduction and adaptation in any medium and for any purpose provided that it is properly attributed. For attribution, the original author(s), title, publication source (PeerJ) and either DOI or URL of the article must be cited.
License URL: https://creativecommons.org/licenses/by/4.0/

Keywords: Apatinib, NSCLC, VEGFR2, Radiosensitivity, AKT, ERK

Funding: National Natural Science Foundation of China 81472806 This work was supported by the National Natural Science Foundation of China (No. 81472806). The funders had no role in study design, data collection and analysis, decision to publish, or preparation of the manuscript.

==============================
Background

Radioresistance is still the major cause of radiotherapy failure and poor prognosis in patients with non-small cell lung cancer (NSCLC). Apatinib (AP) is a highly selective inhibitor of vascular endothelial growth factor receptor 2 (VEGFR2). Whether and how AP affects radiosensitivity in NSCLC remains unknown. The present study aimed to explore the radiosensitization effect of AP in NSCLC and its underlying mechanism as a radiosensitizer.

Methods

The NSCLC cell lines A549 and LK2 were treated with AP, ionizing radiation (IR), or both AP and IR. Expression of VEGFR2 was analyzed by western blot and RT-PCR. Cell proliferation was measured using CCK-8 and colony formation assays. Apoptosis and cell cycle distribution in NSCLC cells were analyzed by flow cytometry. Nuclear phosphorylated histone H2AX foci immunofluorescence staining was performed to evaluate the efficacy of the combination treatment. Western blot was used to explore the potential mechanisms of action.

Results

AP inhibited cell proliferation in a dose- and time-dependent manner. Flow cytometry analysis indicated that AP significantly increased radiation-induced apoptosis. Colony formation assays revealed that AP enhanced the radiosensitivity of NSCLC cells. AP strongly restored radiosensitivity by increasing IR-induced G2/M phase arrest. AP effectively inhibited repair of radiation-induced DNA double-strand breaks. Western blot analysis showed that AP enhanced radiosensitivity by downregulating AKT and extracellular signal-regulated kinase (ERK) signaling.

Conclusion

Our findings suggest that AP may enhance radiosensitivity in NSCLC cells by blocking AKT and ERK signaling. Therefore, AP may be a potential clinical radiotherapy synergist and a novel small-molecule radiosensitizer in NSCLC. Our study fills a gap in the field of anti-angiogenic drugs and radiosensitivity.

Introduction

Lung cancer is the leading cause of morbidity and mortality worldwide, and the incidence and prevalence of lung cancer is increasing (Sung et al., 2021). Approximately 85% of lung cancer patients are diagnosed with non-small cell lung cancer (NSCLC), most of whom are diagnosed at advanced stages. Currently, radiotherapy remains the main therapeutic approach for advanced NSCLC. Although many strategies have been developed to improve the efficacy of radiotherapy for NSCLC, many patients still experience tumor relapse and fatal distant metastasis caused by radioresistance. Therefore, current research focuses on enhancing the radiosensitivity (Kim et al., 2021; Lewis et al., 2021; Seidlitz et al., 2020; Wrona, Dziadziuszko & Jassem, 2021; Xiong et al., 2021). At present, the major types of radiosensitizers include electrophilic radiosensitizers, biological reducers, chemotherapeutic drugs, and natural products (Schrank et al., 2018). Although many radiosensitizers have been studied to improve the radiosensitivity of tumors, their clinical application has not been satisfactory (Ohri et al., 2016). The role of small-molecule targeted drugs in enhancing radiosensitivity in NSCLC is unclear.

Angiogenesis is an essential step in the progression of various types of solid tumors (Chen et al., 2017; Hanahan & Weinberg, 2011). The vascular endothelial growth factor (VEGF) plays a critical role in angiogenesis (Jain, 2005; Jain, 2013). VEGF receptors (VEGFRs) include three protein tyrosine kinases: VEGFR1, VEGFR2, and VEGFR3. VEGFR2 is the main functional tyrosine kinase membrane receptor of VEGF in endothelial cells, playing pivotal roles in transducing VEGF signals and stimulating phosphorylation of extracellular signal-regulated kinase (ERK) through a protein kinase C-dependent pathway. In a variety of malignancies, including NSCLC, the overexpression of VEGF and VEGFR is correlated with increased tumor growth, microvessel density, proliferation, tumor metastasis potential, and poor prognosis. Therefore, inhibiting VEGFR signaling is an attractive therapeutic option in combination with other antitumor regimens in clinical practice (Han et al., 2019; Peng et al., 2017).

Apatinib (AP), a small-molecule inhibitor of VEGFR2, can prevent VEGF-induced phosphorylation of VEGFR2 and subsequent downstream signal transduction, thereby inhibiting the migration and proliferation of endothelial cells stimulated by VEGF. It also inhibits tumor angiogenesis caused by VEGF signaling. AP has been shown to inhibit VEGF-mediated intracellular neovascularization signals in vivo, thereby promoting tumor cell apoptosis and inhibiting cell proliferation. AP alone or in combination with chemotherapy drugs inhibits the growth of several human tumors by inhibiting angiogenesis (Chen et al., 2018; Liu et al., 2017; Procaccio et al., 2019). AP has also been reported to improve radiosensitivity of several human tumors (Hu et al., 2018; Hu et al., 2019; Liang et al., 2020; Liao et al., 2019; Zhao et al., 2017). Yet, whether AP can enhance the radiosensitivity of lung cancer remains to be further explored. We hypothesized that AP may play a novel role in the radiosensitivity of NSCLC cells. Therefore, in the present study, we investigated whether AP enhanced radiosensitivity and explored the underlying mechanisms of radiosensitization in NSCLC cells. The results may provide a theoretical basis for the clinical combination of radiation and AP.

Materials & methods

AP preparation, cell culture, and ionizing radiation (IR) protocol

AP was obtained from the Jiangsu Hengrui Medicine Company (Jiangsu, China). AP was dissolved in 100% dimethyl sulfoxide (DMSO) as a 100 mM stock solution and diluted with RPMI-1640 medium to the desired concentration.

Cells were cultured as previously described in Li et al. (2021). Specifically, the normal mammary epithelial cell line HBEC and four human NSCLC cell lines, A549, H460, H226, and H522, were purchased from the American Type Culture Collection (Manassas, VA, USA). LK2 cells were obtained from the Cell Bank of the Chinese Academy (Shanghai, China). All cells were cultured in RPMI-1640 medium (Sigma-Aldrich, Darmstadt, Germany) and supplemented with 10% fetal bovine serum (Clark Bioscience, Richmond, VA, USA) and 1% penicillin/streptomycin (Sigma-Aldrich, Darmstadt, Germany) in a humidified incubator at 37 °C in 5% CO2. A 6-MeV X-ray medical linear accelerator (Elekta Synergy, Elekta, Stockholm, Sweden) was used to irradiate the cells at a dose rate of 300 cGy/min (dose: 0 to 8 Gy) at room temperature.

RNA extraction, reverse transcription, and qRT-PCR

RNA extraction, reverse transcription, and qRT-PCR were conducted as previously described in Li et al. (2021). Specifically, total RNA was extracted from cells using the TRIzol™ Plus Kit (Takara, Osaka, Japan) according to the manufacturer’s instructions. cDNA was synthesized with total RNA (1 µg) using a real-time PCR system (Life Technologies, Carlsbad, CA, USA), and qRT-PCR was performed with total cDNA (100 ng) using an Applied Biosystems 7,500 Real-Time PCR system with SYBR™ Green Master Mix (Takara, Osaka, Japan). The relative expression of VEGFR2 was determined using the 2−ΔΔCt method after normalization to GAPDH expression. The primers for VEGFR2 were as follows: forward, 5′-GTGATCGGAAATGACACTGGAG-3′ and reverse, 5′-CATGTTGGTCACTAACAGAAGCA-3′.

Flow cytometric analysis

Apoptosis and cell cycle status were conducted as previously described in Li et al. (2021). Specifically, apoptosis was measured using the FITC Annexin V Apoptosis Detection Kit (BD Pharmingen, San Diego, CA, USA) according to the manufacturer’s instructions. Approximately 10 × 105 A549 cells or 15 × 105 LK2 cells were seeded into six-well plates and irradiated with 0 or 8 Gy the next day, cultured for another 48 h, and then incubated with FITC-labeled annexin V and propidium iodide (PI) at room temperature in the dark for 15 min.

Cell cycle analysis was performed using 50 µg/mL PI and 100 µg/mL DNase-free RNase A (Solarbio, Beijing, China). A total of 40 × 105 A549 cells or 60 × 105 LK2 cells were seeded into 25 cm2 Cell Culture Flasks. A total of 24 h post-irradiation, cells were harvested with trypsin, washed with phosphate-buffered saline, and fixed in 70% ice-cold ethanol at 4 °C for 12 h. After washing, the cell pellet was resuspended in PI staining buffer and incubated at 37 °C for 30 min in the dark. Apoptosis and cell cycle status were analyzed by flow cytometry (BD FACSCalibur, BD Biosciences, San Jose, CA, USA).

Cell proliferation and colony formation assay

Cells were seeded into plates with 96 wells at 2,000 A549 cells or 4,000 LK2 cells per well. After exposure to a single dose of radiation (8 Gy), cells were incubated for 24, 48, 72, or 96 h, at which point Cell Counting Kit-8 (Beyotime, Shanghai, China) was used to determine cell viability. Absorbance was measured using a microplate reader (BioTek, Winooski, VT, USA) at a wavelength of 450 nm. The IC50 value of each cell line was calculated using the GraphPad Prism 7 software (La Jolla, CA, USA).

The colony formation assay was conducted as previously described in Li et al. (2021). Specifically, 200 cells were seeded into 6-well plates and irradiated (0, 2, 4, 6, and 8 Gy) the next day. A total of 2 weeks later, the cells were fixed in 4% paraformaldehyde and stained with 0.1% crystal violet, and the number of colonies per dish was counted. The plating efficiency and surviving fraction were calculated as previously described (Sun et al., 2019).

Immunofluorescence detection of phosphorylated histone H2AX (γH2AX)

The immunofluorescence assay was conducted as previously described in Li et al. (2021). Specifically, cells growing on glass coverslips were exposed to 0 or 8 Gy irradiation. A total of 4 h later, the cells were fixed in 4% paraformaldehyde and incubated with the primary phospho-γH2AX antibody (Ser139; Abcam, San Diego, CA, USA) and a secondary antibody conjugated to Cy3 (Beyotime) according to the manufacturer’s protocol. Staining was analyzed using a confocal laser scanning microscope (Nikon, Tokyo, Japan).

Protein extraction and western blotting

Protein extraction and western blotting were conducted as previously described in Li et al. (2021). Specifically, total protein was extracted from cells using radioimmunoprecipitation assay lysis buffer (Beyotime). Total protein concentrations were quantified using a bicinchoninic acid assay kit (Beyotime). Equal amounts (30 µg) of proteins were boiled at 100 °C, separated by sodium dodecyl sulfate-polyacrylamide gel electrophoresis, and transferred onto polyvinylidene fluoride membranes (Millipore, Bedford, MA, USA). The membranes were blocked with 5% (w/v) skim milk at room temperature and immunoblotted overnight at 4 °C with primary antibodies against GAPDH (60004-1-Ig; Proteintech, Chicago, IL, USA), VEGFR2 (abs131800; Absin, Shanghai, China), and AKT (#4685; Cell Signaling Technology, Beverly, MA, USA), phospho-AKT (#4060; Cell Signaling Technology, Beverly, MA, USA), ERK (#4695; Cell Signaling Technology, Beverly, MA, USA), and phospho-ERK (#4370, Cell Signaling Technology, Beverly, MA, USA). The membranes were then incubated with the appropriate horseradish peroxidase-conjugated secondary antibodies (HRP-conjugated goat anti-mouse IgG, Zsgb Bio, ZB-2305; HRP-conjugated goat anti-rabbit IgG, Zsgb Bio, ZB-2301).

Statistical analysis

The experimental data were calculated as the mean ± standard deviation of at least three independent experiments. The data were analyzed, and statistical graphs were created using GraphPad Prism 7. Differences between groups were analyzed using the Student’s t-test, one-way analysis, and two-way ANOVA. Statistically significant differences were determined at p < 0.05 (*), <0.01 (**), or <0.001 (***).

Results

Expression of VEGFR2 in NSCLC cells

The A549, H522, H460, LK2, and H226 cell lines were evaluated for VEGFR2 expression by western blotting and qRT-PCR (Figs. 1A and 1B). VEGFR2 protein expression differed among the five NSCLC cell lines. A549 and LK2 cells were selected for further study based on their expression of VEGFR2.

Figure 1 Expression of VEFGR2 in NSCLC cells.

(A) Differential expression of VEGFR2 protein in different non-small cell lung cancer (NSCLC) cell lines by Western blotting. (B) Quantitative analysis of VEGFR2 mRNA by qRT-PCR. (C) and (E) Apatinib inhibited cell proliferation in a dose and time dependent manner in A549 and LK2 respectively. (D & F) The IC50 value of apatinib after treatment for 72 h in A549 and LK2, respectively. Each data point represents the mean ± SD from three independent experiments. *p < 0.05. **p < 0.01.

To examine the effects of AP on NSCLC cell growth, the cytotoxicity of AP against NSCLC cell lines was determined using the CCK8 assay. The human lung cancer cell lines A549 and LK2 were cultivated with a series of increasing concentrations of AP (0, 5, 10, 20, 40, and 80 μmol/L) for 24, 48, and 72 h. AP significantly inhibited the proliferation of A549 and LK2 cells in vitro in a dose- and time-dependent manner (Figs. 1C and 1E). After calculation, the IC50 doses of AP after treatment for 72 h in LK2 cells and A549 cells were 30.73 μmol/L and 17.61 μmol/L, respectively (Figs. 1D and 1F). We chose the IC20 concentration at 72 h in subsequent experiments.

AP promoted cell death and enhanced radiosensitivity in NSCLC cells

To explore the efficacy of AP in combination with radiotherapy, the CCK8 assay was conducted to determine whether the inhibitory effect of IR was enhanced by AP. LK2 and A549 cells were pretreated with AP or DMSO for 24, 48, 72, or 96 h and exposed to 8 Gy irradiation. Compared to IR alone, cell growth was remarkably suppressed in the cells pretreated with AP before IR at all of the time points examined, although this effect was particularly visible at 96 h. In addition, cells treated with both AP and IR had less proliferation than cells treated with IR or AP alone (Figs. 2A and 2B). These results suggest that AP suppresses the proliferation of NSCLC cells in vitro and enhances the radiosensitivity of NSCLC cells.

Figure 2 Apatinib inhibited the proliferation and enhanced radiosensitivity of NSCLC cells.

(A & B) Cell proliferation was measured using CCK8. (C & D) Apatinib combined irradiation significantly reduced the colony formation ability of A549 and LK2 cells compared with control group, X-ray or apatinib group. Survival fractions obtained by colony formation assays in A549 cells and LK2 cells following treatment with 0, 2, 4, 6, and 8 Gy (right). Each data point represents the mean ± SD from three independent experiments. *p < 0.05. **p < 0.01.

To further confirm the radiosensitivity effects of AP, a colony formation assay was performed. The results revealed that treatment with AP plus IR significantly reduced the survival rate in both NSCLC cell lines compared with cells treated with AP or IR alone. The sensitivity enhancement ratios (SERs) were 1.95 and 2.15 in LK2 and A549 cells, respectively (Table 1). These data suggest that AP sensitizes NSCLC cell lines to IR in vitro.

Table 1 Radiation biological parademeter.

	A549	LK2	
DMSO	Apatinib	DMSO	Apatinib	
k	0.29	0.62	0.28	0.54	
n	1.13	2.46	6.64	8.23	
D0	3.46	1.61	3.6	1.85	
Dq	0.41	1.45	6.81	3.9	
SER	1	2.15	1	1.95	

AP and radiation combination therapy affected apoptosis and cell cycle progression of NSCLC cells

Flow cytometry was used to determine whether AP could induce NSCLC cell apoptosis and enhance the radiosensitivity of A549 and LK2 cells. Annexin V and PI staining were used to detect the percentage of cell death in A549 and LK2 cells treated with AP after 48 h with or without IR. The percentages of apoptotic A549 and LK2 cells were significantly higher in cells treated with AP than in control cells. Further, AP combined with IR remarkably enhanced apoptosis compared with control cells and cells treated with either AP or IR (Fig. 3A). Taken together, these results demonstrate that AP enhances radiation-induced cell apoptosis.

Figure 3 Apatinib promoted radiation-induced apoptosis and induced G2/M phase arrest.

(A) Analysis of cell apoptosis by Annexin-V FITC/PI double staining in A549 and Lk2 cells following treatment with control group, apatinib, X-ray, and combination. (B) Cell cycle analysis in A549 and Lk2 cells following treatment with control group, apatinib, X-ray, and combination. Each data point represents the mean ± SD from three independent experiments. *p < 0.05. **p < 0.01. ***p < 0.001.

To further investigate the effect of AP in NSCLC cells after IR, the cell cycle distribution after treatment was analyzed by flow cytometry. Radiation-induced DNA damage triggers G1 or G2 cell cycle arrest, allowing cells to repair DNA damage. AP has been reported to hamper cell cycle progression, leading to G0/G1 or G2/M arrest. As shown in Fig. 3B, in A549 cells, radiation-induced DNA damage triggered G1 and G2 arrest, and AP induced G2/M arrest. Cell cycle arrest was significantly enhanced by IR. Cell cycle analysis showed that A549 cells treated with both AP and IR had the greatest increase in the G2/M phase proportion, and this proportion was significantly higher than that of control cells (p < 0.01). The same effect was observed in the LK2 cells. Therefore, AP improves the efficiency of IR by causing G2/M arrest. These results indicate that AP enhances radiation-induced apoptosis and G2/M arrest.

Radiosensitization by AP was associated with delayed DNA-double-strand break (DSB) repair

IR induces DNA-DSBs and triggers DNA damage repair responses. The ability to repair DNA-DSBs reflects the radiosensitivity of cells. γ-H2AX was used to monitor the presence of DNA-DSBs to determine whether AP pretreatment enhances radiation-induced DNA damage and interferes with DNA damage repair. Immunofluorescence staining was used to detect the number of γ-H2AX-positive nuclei. There were notably more γ-H2AX-positive cells among cells treated with both AP and IR than in cells treated with IR alone at 4 h post-IR (Figs. 4A and 4B), which reflected inefficient repair of DSBs in cells treated with the combination. These results indicate that AP significantly increases the number of γ-H2AX-positive nuclei and suppresses repair of radiation-induced DNA-DSBs after IR.

Figure 4 Apatinib impaired the ability of radiation-induced DNA-DSBs repair.

(A–B) The representative images of γ-H2AX-positive cells in A549 and LK2 cells treated with IR (8 Gy) or in combination with apatinib at 4 h. γ-H2AX signal in red, nuclear counterstaining with 4′,6-diamidino-2-phenylindole in blue. Scale bar: 50 µm. Values represent the average of three independent experiments (right). *p < 0.05. **p < 0.01. ***p < 0.001.

AP blocks AKT and ERK signaling to sensitize lung cancer cells to radiation

The AKT and ERK pathways are known to be activated by IR; this activation might play a role in radioresistance in NSCLC cells. Our previous results showed that downregulation of the AKT and ERK signaling enhanced radiosensitivity (Sun et al., 2019). In order to determine the potential molecular mechanisms by which AP exerts enhanced antitumor effects and regulates radiosensitivity, we examined the phosphorylation of AKT and ERK by western blotting. The results showed that phosphorylated AKT (p-AKT) and ERK (p-ERK) were reduced after AP treatment in both A549 and LK2 cells. The combination of AP and radiation significantly decreased the activities of AKT and ERK in A549 and LK2 cells (Figs. 5A and 5B). In summary, these data indicate that AP enhances radiosensitivity by decreasing AKT and ERK signaling in NSCLC cells.

Figure 5 Apatinib reduced phosphorylation of AKT and ERK.

(A–B) Protein levels of ERK (p-ERK) and AKT (p-AKT) were detected by western blotting in A549 and LK2 cells. Values represent the average of three independent experiments (right). *p < 0.05. **p < 0.01.

Discussion

Radiotherapy is still one of the most effective measures for the treatment of NSCLC. However, radioresistance contributes to treatment failure. Radiosensitizers have been widely studied in an effort to improve the effectiveness of radiotherapy, but the results have not been satisfactory for NSCLC (Tian et al., 2011). Clinically, the anti-angiogenic drug endostatin exerts a radiosensitizing effect in NSCLC by inhibiting VEGFR2 expression (Liu et al., 2016). Another report suggested that the anti-angiogenic drug bevacizumab enhanced the cytotoxicity of antitumor drugs through the VEGF/VEGFR2 pathway in colon cancer (Liu et al., 2018).

Previous studies have shown that VEGF plays a critical role in angiogenesis in lung cancer, thereby promoting lung cancer progression. Additionally, VEGF overexpression increases the radioresistance of NSCLC, which contributes to disease progression and poor prognosis (Cascone et al., 2017; Chatterjee et al., 2013). VEGFR2 is the main functional tyrosine kinase membrane receptor of VEGF in endothelial cells and plays a pivotal role in transducing VEGF signals. When VEGF is activated, VEGFR2 phosphorylation induces endothelial cell proliferation and migration (Longo & Gasparini, 2007). VEGFR2 is differentially expressed in NSCLC. A tumor cell-autonomous VEGF-VEGFR2 feed-forward loop provides signal amplification required for the establishment of fully angiogenic tumors in lung cancer (Chatterjee et al., 2013). In addition, autocrine regulation of tumor radioresistance occurs through the VEGF-VEGFR2 interaction (Knizetova et al., 2008). AP is an oral small-molecule inhibitor of VEGFR2 that can suppress cell proliferation in a variety of tumors (Feng et al., 2018; Hu et al., 2014; Lu et al., 2017). It has been reported that AP enhances radiosensitivity by inhibiting proliferation in hepatocellular carcinoma and metastatic prostate cancer (Liao et al., 2019; Zhao, Zhang & Qiao, 2017). In this study, we focused on whether AP could be utilized as a novel target drug to enhance radiosensitivity in NSCLC cells and examined the potential mechanism.

It is well known that radiation inhibits tumor growth and promotes apoptosis. Radiation-resistant cancer cells are more capable of surviving radiation and have high proliferative potential. Radioresistant cells may avoid radiation damage by suppressing apoptosis, causing cell cycle arrest, and promoting DNA repair, all of which are related to cancer cell pro-survival signaling pathways following radiation treatment (Hein, Ouellette & Yan, 2014). Our results demonstrated that AP significantly inhibited NSCLC cell proliferation. Moreover, we also observed that the proliferation ability of NSCLC cells was reduced more significantly by AP combined with IR than by either AP or IR alone. In cells treated with both AP and IR, growth inhibition increased as the concentration of AP or the treatment time increased, indicating that this combination treatment works in a dose- and time-dependent manner. To better understand this phenomenon, we performed colony formation assays, which confirmed the radiosensitivity effects of AP. The results revealed that treatment with AP plus IR significantly reduced the survival rate in both NSCLC cell lines compared to treatment with AP or IR alone. The SERs were 1.95 and 2.15 in LK2 and A549 cells, respectively. Collectively, these results led us to postulate that AP might enhance radiosensitivity by decreasing the proliferation and survival abilities of NSCLC cells after IR. Apoptosis is also a key effector mechanism of IR (Hanahan & Weinberg, 2011), and radiotherapy induces and triggers apoptotic cell death. Previous studies have also reported that radiosensitivity may be associated with apoptosis and suppression of angiogenesis in lung cancer cells (Han et al., 2017). Furthermore, a recent study has shown that deregulation of BCL-2 family proteins may overcome radioresistance in NSCLC (Wieczorek et al., 2017). Previous studies showed that AP could promote apoptosis by upregulating the expression of BAX proteins and other pro-apoptotic proteins and downregulating the expression of Bcl-2 in NSCLC cells. Our findings suggested that AP alone could induce tumor cell apoptosis, but combination therapy significantly increased apoptosis compared to treatment with AP or IR alone. Therefore, our data showed that AP may effectively increase radiosensitivity by increasing NSCLC cell apoptosis. Numerous studies have demonstrated that radiosensitization may be attributable to an increase in the percentage of radiosensitive G2/M phase cells (Raju et al., 2002). Recent studies have reported that AP is involved in cell cycle arrest at the G2/M phase via blockade of the cyclin B1/cdc2 complex and upregulation of p21 and p27 in lung cancer cells (Sheng et al., 2018). The present study showed that IR induced G2/M arrest, but, interestingly, there were more cells arrested in the G2/M phase after treatment with both IR and AP. Our observations therefore confirm that AP might increase radiosensitivity by disrupting cell cycle progression and causing cell accumulation in the G2/M phase. Moreover, IR induces DNA-DSBs and DNA damage repair (Clavreul et al., 2018). γ-H2AX is a sensitive marker of DNA-DSBs induced by radiation, is associated with the ability to repair DSBs, and is a marker of radiosensitivity (Yao et al., 2016). Our study found that, in A549 and LK2 cells treated with both AP and IR, the number of γ-H2AX-positive nuclei was significantly higher than that in cells treated with IR alone. Therefore, AP may enhance radiosensitivity by blocking the cell’s ability to repair DNA-DSBs. Taken together, this study revealed for the first time that AP may increase the radiosensitivity of NSCLC cells.

Radioresistant cells have a more malignant phenotype through molecular and genetic alterations, which allows them to survive the cytotoxic effects of IR (Bezjak et al., 2015; Yom, Diehn & Raben, 2015). However, the mechanism underlying radioresistance in NSCLC cells remains unclear. An increasing number of studies have confirmed that inhibiting the AKT and ERK pathways could increase the radiosensitivity of tumor cells in vivo and in vitro (Fumarola et al., 2014; Leszczynska et al., 2015). In our previous study, we found that radioresistant NSCLC cells had activated AKT and ERK and that downregulation of p-AKT and p-ERK could enhance cell radiosensitivity (Sun et al., 2019). Additionally, it has been reported that the PI3K/AKT pathway is involved in blocking apoptosis in lung cancer cell lines; p-AKT increases radioresistance by inhibiting the pro-apoptotic proteins BAD, BAX, and caspase-9 (Toulany & Rodemann, 2015). In addition, the excessive activation of AKT accelerates IR-induced DNA-DSB repair and promotes cell survival post-IR. After DNA damage, DNA-dependent protein kinase is phosphorylated by AKT and accumulates at the damaged site to reinforce the efficiency of DSB repair, which is involved in the mechanism of radioresistance (Toulany et al., 2012). Moreover, ERK-induced radioresistance may inhibit apoptosis by suppressing IR-induced damage to the mitochondrial membrane and increasing the expression of DNA repair proteins (Shonai et al., 2002). Another mechanism by which ERK induces radioresistance is through P90RSK, the downstream effector of ERK, which promotes the expression of a variety of anti-apoptotic proteins by phosphorylating transcription factors (Riccio et al., 1999). Several studies have reported that AP significantly potentiates radiosensitization by promoting apoptosis and inducing cell cycle arrest at the G2/M phase by inhibiting the AKT and ERK pathways, including in cholangiocarcinoma and hepatocellular carcinoma (Huang et al., 2018; Yang & Qin, 2018). In the present study, we observed that IR could activate the AKT and ERK pathways, but AP effectively inhibited the phosphorylation of AKT and ERK. Therefore, AP may increase radiosensitivity by suppressing the activation of AKT and ERK in NSCLC cells. The possible therapeutic benefits of AP in combination with radiation should be further evaluated.

Conclusions

In summary, our study demonstrates that AP enhances the radiosensitivity of NSCLC cells. This occurs by reducing cell proliferation, inducing apoptosis and cell cycle arrest in the G2/M phase, inhibiting DNA-DSB repair, and downregulating AKT and ERK. To our knowledge, this is the first study to demonstrate a novel role of AP as a radiosensitizer in NSCLC. Our findings may provide clinical guidance for improving radiotherapy efficiency in patients with NSCLC. However, whether AP combined with IR is effective for clinical use remains unknown. Therefore, the optimal dose of AP requires further study.

Supplemental Information

Supplemental Information 1 Full-length uncropped blots (Figs. 1 and 5).

Click here for additional data file.

Supplemental Information 2 Raw data: The inhibitory effect of IR was enhanced by AP.

Click here for additional data file.

Supplemental Information 3 Raw data: The cytotoxicity and IC50 of AP.

Click here for additional data file.

Supplemental Information 4 Raw data: colony formation assay.

Click here for additional data file.

Additional Information and Declarations

Competing Interests

Author Contributions

Data Availability

The authors declare that they have no competing interests.

Lin Li performed the experiments, prepared figures and/or tables, and approved the final draft.

Yuexian Li analyzed the data, prepared figures and/or tables, and approved the final draft.

Huawei Zou conceived and designed the experiments, authored or reviewed drafts of the paper, and approved the final draft.

The following information was supplied regarding data availability:

The raw measurements are available in the Supplementary Files.

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
