# Peer review of "A novel role for apatinib in enhancing radiosensitivity in non-small cell lung cancer cells by suppressing the AKT and ERK pathways"

_PeerJ, doi:10.7717/peerj.12356_

## Round 0.1 · original submission · Major Revisions

The current manuscript has several flaws. In terms of presentation please adress the reviewer 1 comments. Reviewer 2 comments on experimental design are also relevant and should be adressed.

Also please cite the following papers, especially the first one, as the results presented in it are a replica of this manuscript. Inclusion of such literature would contextualize better the paper, and instead of presenting it as a novel finding, It would pinpoint its value as a replication of the first paper already mentioned.

Hu, Chenxi et al. “Effects of combined inhibition of STAT3 and VEGFR2 pathways on the radiosensitivity of non-small-cell lung cancer cells.” OncoTargets and therapy vol. 12 933-944. 29 Jan. 2019, doi:10.2147/OTT.S186559

Liang, Li-Jun et al. “Apatinib Combined with Local Irradiation Leads to Systemic Tumor Control via Reversal of Immunosuppressive Tumor Microenvironment in Lung Cancer.” Cancer research and treatment vol. 52,2 (2020): 406-418. doi:10.4143/crt.2019.296

Hu, Chenxi et al. “Role of the NRP-1-mediated VEGFR2-independent pathway on radiation sensitivity of non-small cell lung cancer cells.” Journal of cancer research and clinical oncology vol. 144,7 (2018): 1329-1337. doi:10.1007/s00432-018-2667-8

Zhao, Chunbo et al. “Significant efficacy and well safety of apatinib combined with radiotherapy in NSCLC: Case report.” Medicine vol. 96,50 (2017): e9276. doi:10.1097/MD.0000000000009276

·

Basic reporting

The authors use a clear and unambiguous, professional English throughout the manuscript.

Literature references, sufficient field background/context provided?
1. In the first line of the introduction the authors mention that “Lung cancer is the leading cause of morbidity and mortality worldwide “, and they refer to Siegel et al. 2013, which is a paper on cancer statistics published in 2013. The authors must include a recent reference to support their statement. They could also refer to: https://gco.iarc.fr/. According to this database, Lung cancer is the leading cause of mortality, but the second and forth in incidence and prevalence, respectively.
2. In lines 49-52 the authors mention that the strategies currently used to improve NSCLC are not very efficient, and that “…current research focuses on enhancing the radiosensitivity”, however they cite three references (Cao et al. 2013; Milas et al. 2005; Seidlitz et al. 2020), which are not related to NSCLC, there are many recent reports in the literature reporting different approaches to improve radiosensitivity in NSCLC, it will be better to refer to some of the most recent reports. Some examples are:
• Kim SY, Jeong EH, Lee TG, Kim HR, Kim CH. The Combination of Trametinib, a MEK Inhibitor, and Temsirolimus, an mTOR Inhibitor, Radiosensitizes Lung Cancer Cells. Anticancer Res. 2021 Jun;41(6):2885-2894. doi: 10.21873/anticanres.15070. PMID: 34083279.
• Lewis CD, Singh AK, Hsu FF, Thotala D, Hallahan DE, Kapoor V. Targeting a Radiosensitizing Antibody-Drug Conjugate to a Radiation-Inducible Antigen. Clin Cancer Res. 2021 Jun 1;27(11):3224-3233. doi: 10.1158/1078-0432.CCR-20-1725. PMID: 34074654.

• Xiong L, Tan B, Lei X, Zhang B, Li W, Liu D, Xia T. SIRT6 through PI3K/Akt/mTOR signaling pathway to enhance radiosensitivity of non-Small cell lung cancer and inhibit tumor progression. IUBMB Life. 2021 May 25. doi: 10.1002/iub.2511. Epub ahead of print. PMID: 34033225.
• Wrona A, Dziadziuszko R, Jassem J. Combining radiotherapy with targeted therapies in non-small cell lung cancer: focus on anti-EGFR, anti-ALK and anti- angiogenic agents. Transl Lung Cancer Res. 2021 Apr;10(4):2032-2047. doi: 10.21037/tlcr-20-552. PMID: 34012812; PMCID: PMC8107745.

Professional article structure, figures, tables. Raw data shared.
• The article is presented according to Peerj standards.
• The figures are relevant to the content of the article, and they are in general of sufficient resolution, however the bar graphs in figures 3 and 5 have poor resolution, the resolution must be enhanced.
• The title in Table 1 is “Radiation biological parademeter”, Shouldn’t it be “Radiation biological parameter”?. I suggest including a brief description of the content in Table 1 (the parameters).
• In the graphics in Figure 1 and Figure 2, the authors do not indicate if they performed any analysis to show that the differences are statistically significant. Asterisks should be included to indicate where statistically significant differences were found, and the statistical methods used should be mentioned in the figure legends. They should also indicate if the data are from three or more independent experiments.
• In figure 2 A and B, one of the conditions is labeled as “blank”, were these cells untreated (blank), or were they treated with DMSO? This should be clarified in the labeling and figure legend.
• The same comment as above is for figure 3, one of the conditions in all the graphics is labeled as “blank”. It should be clarified if the cells were untreated or treated with DMSO.
• In figures 3, 4 and 5, the p value for *, ** or *** should be mentioned in the figure legends.
• In figure 4, the label for the Y axes in the bar graphs is “Percentage of positive gH2AX foci cells (%)”. According to the material and methods section what the authors counted was the number of cells showing positive nuclear staining for gH2AX, as far as I understand they didn´t quantified the gH2AX foci. I suggest changing the label to “Percentage of gH2AX positive cells”
• The authors present supplemental materials containing the raw data, however for the colony formation assay they show the images of the assay but there is no excel file with the quantification data. The data for cytotoxicity are presented for the two cell lines in separate excel files, but the data in both excel files are not organized in the same manner making it difficult to verify the data.
• The raw data for ERK WB (LK2) shows only two blots, and in the blot labeled as E1, the pattern of the bands is very different to the pattern in the E2 blot.
• p-AKT blots for A549, there is only one cropped blot shown.
The paper is presented as an appropriate ‘unit of publication’, and it includes all results relevant to the hypothesis.

Experimental design

- The authors clearly define the research question, which is relevant and meaningful. The knowledge gap being investigated is whether AP may play a novel role in the radiosensitivity of NSCLC cells. The authors suggest that the knowledge generated may provide a theoretical basis for the clinical combination of radiation and AP.
Rigorous investigation performed to a high technical & ethical standard.
- The investigation was conducted rigorously and to a high technical standard and in conformity with the prevailing ethical standards in the field.
- Methods are in general described with sufficient information to be reproducible by another investigator. I suggest adding the following information to the methods section:
a) The number of cells that were seeded for proliferation and apoptosis assays
b) The amount of RNA used for cDNA preparation and the amount of cDNA used for qPCR
c) The amount of protein that were used for SDS-PAGE, and the catalog number for the primary antibodies

Validity of the findings

All underlying data have been provided; they are robust, statistically sound, & controlled.
The conclusions are appropriately stated, they connected to the original question investigated, and are supported by the results. However. I suggest modifying the sentence “accelerating cell cycle arrest in the
329 G2/M phase”, as the authors didn´t evaluate the speed of the cell cycle arrest. The experiments in figure demonstrate that there is an increase in the number of cells arrested at G2/M, but there is no evidence that this occurs because the arrest is accelerated.

·

Basic reporting

no comment

Experimental design

no comment

Validity of the findings

no comment

Additional comments

A novel role for apatinib in enhancing radiosensitivity in non-small cell lung cancer cells by suppressing the AKT and ERK pathways by Li, L, et al. show results of the radiosensitive effect of apatinib on NSCLC cell lines. The results are interesting, however, authors should respond to some questions.

1. Why the authors didn´t use H226 cell line to evaluate the effect of apatinib and radiotherapy? H226 cell line express higher levels of VEGFR2. The effect of apatinib must be evaluated in H226 cells.

2. LK2 cells express lower levels of VEGFR2 (almost imperceptible), these cells must represent the negative control of the effect of apatinib, however, LK2 cells behavior are similar to A549 cells, in fact, LK2 cells are more sensitive to apatinib and radiotherapy (LK2 cells showed higher levels of apoptosis and lower clonogenic survival). Radiotherapy is able to induce the expression of several membranal receptors to promote survival, its important that authors evaluate the expression of VEGFR2 after treatment of radiotherapy, apatinib or both (as figure 5) to explain the similar results between both cells. These results must explain the effect of apatinib on NSCLC cells.

---

## Round 0.2 · accepted · Accept

Thank you for addressing reviewers' critiques and for amending your manuscript. I am pleased to accept the revised version in its present form.

·

Basic reporting

i have no comment

Experimental design

i have no comment

Validity of the findings

i have no comment

Additional comments

i have no comment